# Correlation of Transcriptomics and FDG-PET SUVmax Indicates Reciprocal Expression of Stemness-Related Transcription Factor and Neuropeptide Signaling Pathways in Glucose Metabolism of Ewing Sarcoma

**DOI:** 10.3390/cancers14235999

**Published:** 2022-12-05

**Authors:** Carolin Prexler, Marie Sophie Knape, Janina Erlewein-Schweizer, Wolfgang Roll, Katja Specht, Klaus Woertler, Wilko Weichert, Irene von Luettichau, Claudia Rossig, Julia Hauer, Guenther H. S. Richter, Wolfgang Weber, Stefan Burdach

**Affiliations:** 1Department of Pediatrics and Children’s Cancer Research Center, Kinderklinik München Schwabing, Klinikum Rechts der Isar, Fakultät für Medizin, Technische Universität München, 80804 Munich, Germany; 2Institute of Pathology, Technische Universität München, 81675 Munich, Germany; 3Department of Nuclear Medicine, University Hospital Münster, Albert-Schweitzer-Campus 1 A1, 48149 Munster, Germany; 4Musculoskeletal Radiology Section, Klinikum Rechts der Isar, Technische Universität München, 81675 Munich, Germany; 5German Cancer Consortium (DKTK), Partner Site Munich, 81675 Munich, Germany; 6ERN PaedCan, 1090 Vienna, Austria; 7Department of Pediatric Hematology and Oncology, University Children’s Hospital Muenster, 48149 Muenster, Germany; 8Cells-in-Motion Cluster of Excellence (EXC 1003-CiM), University of Muenster, 48149 Muenster, Germany; 9Department of Pediatrics, Division of Oncology and Hematology, Charite–Universitätsmedizin Berlin, Augustenburger Platz 1, 13353 Berlin, Germany; 10German Cancer Consortium (DKTK), Partner Site Berlin, 13353 Berlin, Germany; 11Department of Nuclear Medicine, Klinikum Rechts der Isar, Technische Universität München, 81675 Munich, Germany; 12Academy of Translational Medicine and Department of Molecular Oncology–British Columbia Cancer Research Centre, University of British Columbia, 675 West 10th Avenue, Vancouver, BC V5Z 1L3, Canada

**Keywords:** Ewing sarcoma, 18-FDG-PET, SUVmax, radiogenomics, transcriptomics

## Abstract

**Simple Summary:**

Survival rates for metastatic or early recurring Ewing sarcoma (EwS) are dismal, and therapies have severe side and long-term effects. Both aspects require new therapeutic options and targets for treatment. Risk stratification enables individualized treatment and may reduce the burden of side effects. Our radiogenomics study provides novel candidates at the gene expression level to explain the mechanisms of malignancy. We retrospectively analyzed 19 EwS samples (17 patients) and integrated functional genomics assessed by gene expression and functional imaging assessed by FDG-PET, which has the potential to better characterize these highly malignant tumors. The identified genes and pathways can serve as a starting point for prospective experimental and clinical studies of new therapeutic interventions. Thus, this study opens new opportunities for future studies to improve the outcome of patients with poor survival rates.

**Abstract:**

Background: In Ewing sarcoma (EwS), long-term treatment effects and poor survival rates for relapsed or metastatic cases require individualization of therapy and the discovery of new treatment methods. Tumor glucose metabolic activity varies significantly between patients, and FDG-PET signals have been proposed as prognostic factors. However, the biological basis for the generally elevated but variable glucose metabolism in EwS is not well understood. Methods: We retrospectively included 19 EwS samples (17 patients). Affymetrix gene expression was correlated with maximal standardized uptake value (SUVmax) using machine learning, linear regression modelling, and gene set enrichment analyses for functional annotation. Results: Expression of five genes correlated (*MYBL2*, *ELOVL2*, *NETO2*) or anticorrelated (*FAXDC2*, *PLSCR4*) significantly with SUVmax (adjusted *p*-value ≤ 0.05). Additionally, we identified 23 genes with large SUVmax effect size, which were significantly enriched for “neuropeptide Y receptor activity (GO:0004983)” (adjusted *p*-value = 0.0007). The expression of the members of this signaling pathway (*NPY*, *NPY1R*, *NPY5R*) anticorrelated with SUVmax. In contrast, three transcription factors associated with maintaining stemness displayed enrichment of their target genes with higher SUVmax: *RNF2*, E2F family, and *TCF3*. Conclusion: Our large-scale analysis examined comprehensively the correlations between transcriptomics and tumor glucose utilization. Based on our findings, we hypothesize that stemness may be associated with increased glucose uptake, whereas neuroectodermal differentiation may anticorrelate with glucose uptake.

## 1. Introduction

Ewing sarcoma (EwS) is a malignancy of bone or soft tissue. Incidence and prognosis decrease with age [1,2]. Survival rates have increased from 10% in the 1970s to 70–80% in 2000 for localized disease [1,3], due to multimodal approaches combining chemotherapy, surgery, and radiation. However, the 5-year survival rate for patients with advanced stages is still <30% [1,3]. Little therapeutic progress has been made in the last two decades [4,5,6]. Stratification of therapeutic intensity is critical in children because of long-term effects [1]. Toxicity and poor survival rates require individualization of therapy with new approaches [7]. Better mechanistic understanding may help to address these issues.

Resistance to targeted therapies is related to heterogeneity and plasticity of tumors [8]. Biopsy of a single lesion is the diagnostic gold standard. However, the complexity of a systemic disease cannot be mapped with such a locally restricted procedure. Whole-body molecular imaging might aid in overcoming these limitations. Radiomics offers new opportunities for analyzing these large imaging datasets. The combination of advanced image analysis with tissue-based genomic data—called radiogenomics—allows an in-depth characterization of the disease.

EwS cells show a stem cell-like phenotype with poor differentiation [9,10]. The exact cell of origin is still under debate [9,11,12,13,14,15]. Genetically, EwS is uniformly characterized [1,2,10,16,17,18]. Epigenetics and transcriptional regulation are thought to account for variability [1,2,9,16,19]. However, it is unclear how they relate to imaging.

18-F-fluorodeoxyglucose positron emission tomography (18-F-FDG PET) is a functional imaging method that quantitatively measures glucose uptake. A common parameter is the maximum standardized uptake value (SUVmax), the highest rate of FDG uptake in the tumor indicating metabolic tumor activity. Quantitative FDG-PET parameters, and in particular, SUVmax, have been shown to be potentially prognostic in several cancers, both in primary and recurrence, pre-treatment and post neoadjuvant therapy, correlating with tumor growth, worse survival, poor prognosis, advanced stage, and worse course of disease [20,21,22,23,24,25,26,27,28,29,30,31,32,33,34,35]. SUVmax correlates with worse outcome and stage of the disease in primary EwS [36,37,38,39,40,41] while studied less in recurrence. However, primary and relapsed cases show similar SUVmax [42]. To further investigate the role of SUVmax in EwS, we use it as a quantitative phenotype in our study.

The aim of the present study was to characterize variable glucose uptake in EwS by identifying correlations of SUVmax and transcriptomics. These correlations indicate which genes and pathways may be more or less active in tumors with regard to SUVmax. Given the potential prognostic value of SUVmax suggested by the literature, we hypothesize that such genes or pathways may help improve our mechanistic understanding and qualify for experimental validation.

## 2. Materials and Methods

### 2.1. Dataset: The Munich Cohort (n = 19)

We included all patients suffering from EwS in the Children’s Hospital Schwabing and in the Klinikum rechts der Isar of the Technical University of Munich in the years 2011 to 2019 who fulfilled our inclusion criteria. The criteria were: all patients suffering from primary or relapsed EwS, aged up to 40 years, with image data and tissue sampled from the same lesion. Patients with pre-therapeutic primary disease or pre-therapeutic relapse were classified as “untreated”, otherwise “under therapy”, at the time of PET imaging or tissue sampling. The time interval between tissue sampling and imaging had to be maximum 6 weeks for untreated patients, and maximum 2 weeks for patients under therapy, thus ensuring that tissue sample and PET reflected the same biological characteristics of the tumor. This yielded 19 samples from 17 patients (Table 1, Appendix A). From 2 patients, 2 samples each were included, representing different lesions at different time points in the disease course.

The data were analyzed retrospectively. The registry study was approved by the local ethics committee (reference 223/16S) and complies with the Declaration of Helsinki. All patients gave their written consent.

### 2.2. Gene Expression Data

#### 2.2.1. Tissue

Frozen tumor samples for expression analysis were obtained from biopsies or resection specimens and passed the quality control of experienced pathologists. Sample preparation followed the Affymetrix protocol and was previously described [12]. Gene expression was measured using [HuGene-1_0-st] Affymetrix Human Gene 1.0 ST Array.

#### 2.2.2. Preprocessing

The microarray data was processed using Robust Multichip Average (RMA) preprocessing [43,44,45] (R package oligo [46]), including background subtraction, quantile normalization, and median-polish summarization of probe sets to genes (log2 expression values). For summarization, Brainarray (version 24) [47,48,49] was used, yielding 20,722 genes.

#### 2.2.3. Filtering

We applied two filtering steps on our gene expression data. In the first step, we excluded genes with low expression (average expression below 10), as microarray chips are not precise at low expression levels [50]. A total of 20,524/20,722 genes remained.

For the second filtering step, we wanted to focus on genes that might be related to survival in EwS. To obtain such “potential survival genes”, we applied a machine learning approach to external datasets. We collected all EwS datasets from the Gene Expression Omnibus (GEO) database [51] (as of March 2018) of gene expression plus survival data, which yielded 3 datasets: GSE63155 [52], GSE17618 [53], GSE63156 [52]. The survival data were used to split the patients into long-term survival (overall survival, OS ≥ 5 years) or short-term survival (OS < 5 years). Ambiguous patients (i.e., patients lost to follow-up within 5 years) were excluded, as it is unknown whether they died before or after the 5-year cutoff. This yielded n = 31 for GSE63155 (12 short OS, 19 long OS), n = 40 for GSE17618 (22 short OS, 18 long OS), and n = 31 for GSE63156 (9 short OS, 22 long OS). Each dataset was analyzed separately to avoid bias of different data sources. These external expression data were preprocessed in the same way as our expression data described above. To reduce dimensionality, 50% of the genes with low coefficient of variation were removed, as these were supposed to be not useful for the discrimination task. Random forest classification was applied to each dataset to predict binary OS: short OS vs. long OS (R package caret [54], method “rf” using 1000 trees). Due to class imbalance in all datasets, downsampling to equal class sizes was performed to avoid bias for one class. Then, random forest models were generated for each dataset in terms of repeated 10-fold cross-validation. Each model randomly chose genes and from this subset selected the gene that best discriminated between samples with long and short OS. This gene was incorporated into a decision tree. This way, each model built 1000 decision trees, whose ensemble vote would be used for classifying a new, unseen sample as “long OS” or “short OS”. The genes, which were used in the decision trees, contributed information to the discrimination task. This way, the model identified a set of genes that were informative for predicting survival time in each dataset. We called the overlap of the 3 gene sets “potential survival genes” (1491 genes). Functional annotation of these genes was obtained using DAVID Bioinformatics Resources (v6.8) [55,56], category UP_KEYWORDS. Adjusted *p*-values < 0.01 (Benjamini) were considered significant. The “potential survival genes” were used to filter the expression data in our cohort (1376/1491 genes, Appendix A).

### 2.3. Imaging Data

We analyzed PET computed tomography (PET-CT)/PET magnetic resonance (PET-MR) to obtain SUVmax for those lesions that were used for gene expression measurements. All patients underwent 18F-FDG PET imaging for diagnostic or staging purposes in our institution.

The PETs were controlled for quality using OsiriX DICOM viewer [57], and tumors were delineated using cuboids by experienced nuclear medicine physicians. Exact delineations were obtained based on a standardized uptake value (SUV) threshold of 40% of SUVmax for each lesion. Image features were calculated according to the image biomarker standardization initiative (IBSI) [58] using PyRadiomics [59] v3.0.1 with standard settings (Python version 3.8 as of October 2020). Spatial resampling to 4 × 4 × 4 mm was applied. For image discretization, a fixed bin size of 0.5 was used. SUVmax was obtained from the original image, i.e., no filters were applied (PyRadiomics feature “original_firstorder_Maximum”).

### 2.4. Statistical Analysis of SUVmax and Clinical Variables

#### 2.4.1. Distribution of SUVmax Regarding Clinical Variables

We tested for equal distribution of SUVmax values regarding the clinical variables: sex (male, female), disease state (primary disease, relapse), sample type (tumor, metastasis), therapy (under therapy, untreated), and age (≤15 years, >15 years). A two-sided Welch two-sample t-test was applied (R function stats::t.test). *p*-values < 0.05 were considered significant.

#### 2.4.2. Survival Analysis

We applied univariate Kaplan–Meier analyses for OS with log-rank tests (R package survival [60,61], functions survfit and ggsurvplot). We tested sex, disease state, age, and SUV categories (samples split by median SUVmax into low SUV vs. high SUV).

We built multivariate Cox proportional hazards models (R package survival [60,61], function coxph), including continuous SUVmax and disease state. A forest plot was generated using R package survminer [62], function ggforest. *p*-values < 0.05 were considered significant.

### 2.5. Correlation Analysis

#### 2.5.1. Linear Regression

We used linear regression (“least squares”) to correlate gene expression with SUVmax. Moderated t-statistics were calculated based on an empirical Bayes method in the R package limma [63]. We applied the limma-trend method to fit a trend to the prior variances. Limma was run on all genes with average expression above 10 (20,524 genes). Afterwards, the model results were filtered for “potential survival genes” (1376 genes). Adjusted *p*-values < 0.05 (Benjamini–Hochberg) were considered significant.

In general, when comparing expression between two groups, a doubling of gene expression is usually considered to be of interest. When using fold changes on the log2 expression values (logFC), a doubling of gene expression corresponds to a logFC cutoff of 1. For the linear regression used in this analysis, we defined relations of high effect size by transferring the standard abs(logFC) cutoff of 1 to regression modeling, which corresponded to abs(slope) >0.146. A total of 23 genes fulfilled this criterion.

The heatmap depicting Z-scaled gene expression in all samples with clinical data in the side bars was generated using R function GMD::heatmap.3 [64]. The dendrograms are based on Euclidean distance and average linkage.

#### 2.5.2. Enrichment Analyses

The results of linear regression were further tested in enrichment analyses for functional annotation by applying 2 tools: Enrichr [65,66,67] and gene set enrichment analysis (GSEA v4.1.0) [68,69]. The set of 23 genes of high effect size was analyzed using Enrichr (applied in September 2021). We focused on pathways in “Reactome 2016” and the Gene Ontology (GO) knowledgebase, including “GO biological process 2021”, “GO molecular function 2021” and “GO cellular component 2021”. Adjusted *p*-values < 0.01 were considered significant.

Additionally, we performed GSEA based on all 1376 genes ranked by their correlation with SUVmax (decreasing slope). For each gene set, a normalized enrichment score (NES) is calculated, which indicates the extent to which this gene set is enriched at the top or bottom of the ranked gene list. *p*-values are corrected for multiple testing (false discovery rate, FDR, q-value). If GSEA is used for hypothesis generation, the developers suggest a less stringent cutoff for significance. Thus, FDR < 0.1 was considered significant.

## 3. Results

An overview of all analysis steps is provided in Figure 1. In general, we used R Statistical Software (v4.0.2) [70] for data analysis, statistical testing, and plot generation.

### 3.1. The Munich Cohort (n = 19)

Out of 75 patients referred to our institution during the duration of the study, only 17 (Table 1, Appendix A) met all our quality standards. We included 19 samples (primary disease n = 5, recurrence n = 14) from 17 EwS patients (female n = 10, male n = 7), aged three to thirty-one years. Tissue samples for expression analysis were obtained from one lesion each (tumor n = 5, metastasis n = 14) before (n = 12) or during (n = 7) treatment. From this tissue, expression of all genes was assessed by Affymetrix Gene Chip analysis. Of the same lesion, we also analyzed PET-CT (n = 15)/PET-MR (n = 4) to obtain SUVmax. The glucose uptake varied in our cohort (Figure 2a). SUVmax showed a spectrum of 1.9 to 21.3, with a median of 5.4. Based on the median, the samples were split into two groups: lesions with low SUV (n = 9, SUVmax 1.9 to 5.1, median 2.8) and lesions with high SUV (n = 10, SUVmax 5.4 to 21.3, median 9.0).

### 3.2. Statistical Analysis of SUVmax and Clinical Variables

#### 3.2.1. SUVmax Distribution with Regard to Clinical Variables

We explored the correlation of SUVmax with clinical variables in our cohort. Potential correlations would render the clinical variables confounding factors and introduce bias in subsequent analyses, in which we correlated gene expression with SUVmax. We found no significant difference in SUVmax distribution with respect to sex, disease state, sample type, therapy, and age (Figure 2b).

#### 3.2.2. Survival Analysis

Univariate Kaplan–Meier analysis (Appendix A) indicated no significant correlation of clinical variables (sex, disease state, and age) or SUVmax with OS (all *p*-values > 0.05). To investigate continuous SUVmax and incorporate more variables, a multivariate Cox proportional hazards model was built in the next step. We used SUVmax and added disease state, as it has shown a trend in Kaplan–Meier analysis (*p*-value = 0.09). In combination, both variables showed significant correlation with survival (overall *p*-value = 0.02 in log-rank test, Appendix A): higher SUVmax (*p* = 0.02, hazard ratio, HR, [95% confidence interval (CI)] = 1.2 [1.0; 1.3]), and relapse (*p* = 0.05, HR [95% CI] = 5.0 [1.0; 24.9]) were associated with increased risk of death.

### 3.3. Gene Expression Analysis

We analyzed the expression of 20,722 genes passing preprocessing and quality control (Figure 1, right panel). To overcome the problem of high dimensionality in such large-scale analyses, we performed two filtering steps. First, we excluded genes with low expression. In the second filtering step, we focused on genes that are potentially linked to survival in EwS. To obtain such genes, we used a machine learning approach on three external, public EwS datasets: GSE63155, GSE17618, GSE63156 (Figure 3a). The model identified 1491 genes that were predictive in all three external datasets, denoted as “potential survival genes”. Functional annotation of these genes yielded phosphoprotein, alternative splicing, polymorphism, acetylation, cytoplasm, cell division, cell cycle, Golgi apparatus, DNA replication, disease mutation, mitosis, cell junction, and endoplasmic reticulum (Figure 3b). We focused on the “potential survival genes” for the correlation analysis with SUVmax. After both filtering steps, we obtained a total of 1376 genes, whose log2 expression values were used for further analysis.

### 3.4. Correlation Analysis of SUVmax and Gene Expression

#### 3.4.1. Linear Regression

To investigate correlations of gene expression and SUVmax (Figure 1, lower panel), we applied linear regression modeling for all 1376 genes that remained after the preprocessing and filtering steps. The volcano plot (Figure 4a) illustrates the effect size (i.e., the slope of the regression line) and r^2^. For 645/1376 genes, the regression slope was positive, for 731/1376 genes, the regression slope was negative. A volcano plot depicting slopes and adjusted *p*-values is supplied in the Appendix A.

#### 3.4.2. Significant Association of SUVmax and Five Genes

A total of 5/1376 genes showed a significant correlation with SUVmax (adjusted *p* < 0.05). Of these, 3/5 genes were positively associated with SUVmax (Figure 4b, top row): *MYBL2* showed a slope of the regression line of 0.149 (95% CI [0.088; 0.211]), indicating that the gene expression doubled over 6.69 SUV units (r^2^ = 0.58). *ELOVL2* showed a slope of 0.148 (95% CI [0.084; 0.212]), indicating that the expression doubled over 6.76 SUV units (r^2^ = 0.54). *NETO2* showed a slope of 0.157 (95% CI [0.086; 0.228]), indicating that the expression doubled over 6.38 SUV units (r^2^ = 0.51). In contrast, 2/5 genes were negatively associated with SUVmax (Figure 4b, bottom row): *FAXDC2* showed a slope of −0.176 (95% CI [−0.246; −0.107]), indicating that the expression halved over 5.67 SUV units (r^2^ = 0.59). *PLSCR4* showed a slope of −0.223 (95% CI [−0.321; −0.125]), indicating that the expression halved over 4.49 SUV units (r^2^ = 0.51).

#### 3.4.3. 23 Genes with High Effect Size

Next, we examined genes correlating with SUVmax with high effect size. Effect size is important, i.e., how much gene expression changes in relation to SUVmax. Genes with a high difference in expression levels in relation to SUVmax are more likely to have an impact of biological relevance. To define genes with high effect size, we chose a cutoff for the slope: abs(slope) > 0.146. Thereby, we obtained 23 genes with high effect size (Figure 4a labelled genes, Appendix A). The majority of genes was negatively correlated with SUVmax (17/23), 6/23 positively. The 23 genes included the five significant correlations, so the previously described five genes had a high effect size by our definition.

The expression of the 23 genes in the 19 samples is illustrated in a heatmap (Figure 5a), together with clinical variables and a stratification into two groups of low SUV or high SUV according to median SUVmax. Visual inspection of the hierarchical clustering of the samples based on expression showed that the two tumors with the highest glucose uptake in our cohort (SUVmax 13.6 and 21.3) and two tumors with low SUVmax (2.7 and 4.3) clustered apart from the other samples. The remaining samples split into two groups: a cluster with lower SUVmax (5/6 in low SUV group), and a cluster with higher SUVmax (7/9 in high SUV group). As this sample clustering was based on genes whose expression changed strongly in relation to SUVmax, it reflected the samples’ spectrum of SUVmax values. This suggested that there was an expression signature for metabolic activity.

The hierarchical clustering of the genes displayed two clusters. In the smaller cluster, 5/6 genes correlated positively with SUVmax. In the larger cluster, 16/17 genes correlated negatively with SUVmax. Thus, the clustering reflected the two directions of association with SUVmax.

### 3.5. Enrichment Analyses

After analyzing single correlations of gene expression and SUVmax, we looked for shared pathways and processes that correlated with glucose uptake. The aim was to summarize and generalize the results of the correlation analysis on a functional level. We investigated whether there were pathways and annotated gene sets distinguishing tumors with varying glucose uptake by applying different approaches of enrichment analysis. Enrichment analyses are robust to false positive findings because many genes are considered at once. This is an advantage, especially with a small sample size.

#### 3.5.1. Enrichments among the 23 Genes with High Effect Size

First, we examined the functional annotation of genes with high effect size. For the 23 genes that correlated with SUVmax with high effect size, we scanned their annotation for their prognostic value in cancer entities (Appendix A), and then performed enrichment analysis for shared pathways and function.

According to the Human Protein Atlas [71], 14/23 genes are related with survival in several cancer types. High expression of *ABCA5*, *C5*, *DNASE1L3*, *ELOVL2*, *FAXDC2*, *NPY1R*, *SLC38A4*, *TES,* and *ZDHHC21* predicts a favorable outcome in breast, liver, pancreatic, renal, and urothelial cancer. In contrast, high expression of *FRZB, MYBL2, MYL2, NETO2, PLSCR4,* and *TES* predicts an unfavorable outcome in endometrial, head and neck, liver, pancreatic, and renal cancer.

We tested for enrichment of pathways and functions systematically using the tool Enrichr [65,66,67]. The set of 23 genes with high effect size showed significant enrichment for the Reactome pathway “Peptide ligand-binding receptors Homo sapiens R-HAS-375276” with q = 0.0043 (Figure 5b, top), which included a subset of the rhodopsin-like G protein-coupled receptor (GPCR) family. This enrichment was due to four genes: *C5, NPY1R, NPY5R,* and *GRP*. There were no significant enrichments in GO biological process and GO cellular component. However, the 23 genes were significantly enriched for the GO molecular function “neuropeptide Y receptor activity (GO:0004983)” with q-value = 0.0007 (Figure 5b, bottom). NPY receptors are also rhodopsin-like receptors. This enrichment was based on the two genes, *NPY1R* and *NPY5R*.

As the NPY receptors contributed to both enrichments, we further investigated the role of the NPY pathway in our dataset. The expression of the genes in the NPY signaling axis was decreased with increasing SUVmax (Appendix A). The signaling molecule *NPY* showed a slope of the regression line of −0.136 (95% CI [−0.263; −0.009]), implying expression halved per 7.37 SUV units (r^2^ = 0.18). The NPY receptor *NPY1R* showed a slope of −0.334 (95% CI [−0.514; −0.155]), indicating expression halved per 2.99 SUV units (r^2^ = 0.40). *NPY5R*, another receptor, showed a slope of −0.222 (95% CI [−0.433; −0.010]), that is, expression halved per 4.51 SUV units (r^2^ = 0.17). In contrast, two other NPY receptors (*NPY2R* and pseudogene *NPY6R*) were expressed constantly, regardless of SUVmax. *NPY2R* showed a slope of 0.00007 (95% CI [−0.043; 0.043]; r^2^ = 0.00), and *NPY6R* showed a slope of 0.009 (95% CI [−0.030; 0.048]; r^2^ = 0.01). In addition, there were two paralogs of *NPY*, namely, *PYY* and *PPY*. Their expression was independent of SUVmax as well: *PYY* showed a slope of −0.005 (95% CI [−0.056;0.046]; r^2^ = 0.00), and *PPY* showed a slope of 0.015 (95% CI [−0.025; 0.054]; r^2^ = 0.03).

All in all, of the 23 genes that were strongly associated with SUVmax, most are associated with survival in several cancer entities, and therefore play a role in distinguishing subgroups in entities other than EwS. Our findings, namely, that these genes have different expression levels in EwS tumors with high or low glucose uptake potentially indicating a different prognosis, suggest that they may also distinguish subgroups in EwS. Looking for functional similarities of the 23 genes, we found that the NPY signaling axis seems to correlate negatively with glucose uptake.

#### 3.5.2. GSEA Based on Regression Results of All 1376 Genes

In addition to enrichments in the set of 23 genes with high effect size, we investigated enrichments across the entire results obtained from linear regression. To this end, we used GSEA, which has several advantages compared to enrichment methods working on a set of genes of interest. First, no cutoff has to be chosen to determine the gene set for enrichment testing, which makes the whole analysis less arbitrary. Second, GSEA utilizes much more information because the entire linear regression results serve as input, namely, a list of genes ranked by their correlation with SUVmax. Using this ranking of genes, the direction of the association is considered: whether enrichment for a gene set occurs at the top of the list (among genes that correlate positively with SUVmax) or at the bottom (among genes that correlate negatively with SUVmax). Thus, GSEA provides a broader view of which functional gene sets are related to glucose uptake, compared to the analysis limited to the 23 genes with high effect size.

With GSEA, we tested different categories of gene sets. First, we used “hallmark gene sets” (H), which provided initial insight and a general overview of all categories. We then focused on more specific aspects. To investigate pathways, we used “curated gene sets: canonical pathway” (C2cp), which summarized pathways from five databases (BIOCARTA, KEGG, PID, REACTOME, and WikiPathways). We also scanned for transcription factors (TFs) whose targets were positively or negatively correlated with SUVmax. Thus, we used “regulatory target gene sets: TF targets” (C3tft), which contained gene sets that shared TF binding sites or motifs. Finally, we tested “ontology gene sets” (C5) containing terms from GO and the Human Phenotype Ontology (HPO).

The results of GSEA—the number of enriched terms found for each category—is listed in Appendix A. Across all categories, we observed more significant enrichments among genes with positive regression slope, i.e., genes upregulated in tumors with higher SUVmax, than enrichments among genes with negative regression slope, i.e., genes upregulated in tumors with lower SUVmax. In H, there were three terms enriched among genes with positive slope, in C2cp there were 16 terms, and in C5 there were 71 terms—all representing basic functions such as cell cycle, DNA replication and repair, transcription, cytoskeleton, actin–myosin interaction, muscle, and muscle development. These processes seem to be upregulated in EwS cells relative to increased glucose uptake and reflect raised metabolic activity.

However, we also found enrichments that cover more specific aspects. Looking at category C2cp for canonical pathways, two pathways were significantly enriched among genes with negative slope (Figure 6a): “REACTOME_PEPTIDE_LIGAND_BINDING_RECEPTORS” (NES −1.90; q = 0.042) and “WP_GPCRS_CLASS_A_RHODOPSINLIKE” (NES −1.86; q = 0.036), which both involve rhodopsin-like GPCRs. For both terms, the enrichments were predominantly due to genes of the NPY signaling axis (so-called core enrichment, Appendix A). These findings mirrored the results of the previous enrichment analysis of Enrichr on the 23 genes with high effect size.

Furthermore, in category C3tft of TF targets, three terms showed significant enrichment. The targets of *RNF2* (NES 2.26; q = 0.003), of the E2F family of TFs (NES 2.00; q = 0.065), and of *TCF3* (NES 1.99; q = 0.048) all showed enrichment among genes positively associated with SUVmax (Figure 6b, Appendix A). This may indicate that the activity of *RNF2*, the E2F family, and *TCF3* was related to the glucose uptake by EwS cells.

In summary, we identified pathways and TFs that characterized tumors in terms of glucose uptake. Tumors with higher glucose uptake had more terms than tumors with lower glucose uptake, especially terms referring to increased turnover, such as cell cycle, replication, and transcription. This may also reflect the increased glucose uptake. Furthermore, the activity of the three TFs *RNF2*, the E2F family, and *TCF3* might be positively associated with glucose uptake, whereas rhodopsin-like receptor pathways might be negatively associated with glucose uptake in EwS tumors.

## 4. Discussion

Our large-scale analysis comprehensively examined the correlations between gene expression and variable glucose uptake from PET. EwS has long been known for its predilection for glucose utilization and FDG-PET sensitivity [72], and inhibiting glycolysis was promising [73]. We found genes, signaling pathways, and TFs characterizing EwS tumors in terms of SUVmax, i.e., glucose uptake and consecutive glycolysis.

Most studies investigating the potential prognostic value of PET have in common that high SUV values indicate a more aggressive stage of disease and worse survival, although the exact cutoff values vary. Nevertheless, it is established that higher glucose uptake indicates malignancy, in general associated with the Warburg effect [74,75]. As PET signals such as SUVmax are supposed to be prognostic in EwS, we anticipate that our results provide novel insight to explain the mechanisms of malignancy, and potentially reveal new therapeutic options in the future.

### 4.1. Limitations

Due to the low prevalence, studies of EwS have a small sample size [2]. As the radiogenomics approach is not established for pediatric sarcoma yet, public datasets of imaging and expression data are not available. However, we are presently validating our findings on an external dataset.

Another limitation is the heterogeneity of clinical variables. There are multiple prognostic factors even in EwS, such as age, ethnic background, localized or metastatic disease, primary site, tumor volume, response to therapy, and primary disease or recurrence, in which the time to recurrence also has an influence [76]. With this complex interplay of risk factors, it is futile to stratify patients just by one risk factor. Limited sample sizes do not allow to create subgroups with respect to multiple risk factors. To compensate for this, we checked for SUVmax distribution with respect to risk factors that were available for our dataset. Among the clinical variables, we found no significant differences in SUVmax distribution, suggesting that single risk factors do not introduce an obvious bias into our data.

A third limitation is the skewed SUVmax distribution in our cohort. The sample with a very high SUVmax of 21.3 (sample_19) may influence the linear regression more than one of the other samples with SUVmax between 1.9 to 13.6. To evaluate the sensitivity of our findings towards sample_19, we compared the results from regression analysis with and without sample_19. We conclude that the sample with high SUVmax has an impact on the correlation results in some cases. The significant results rely on the presence of sample_19, while the findings with high effect size are mostly independent of it. However, there is no reason to consider that the results based on the 18 samples are more correct than the results based on the 19 samples. For completeness, we have pointed out the problem that might be caused by the skewed SUVmax distribution in our cohort. To dispel doubts and to validate the results, we propose to repeat this analysis on a larger dataset with a better-covered range of SUVmax values.

As a consequence of the small, heterogeneous cohort, we focused on the results of robust methods, such as functional enrichment analyses. Based on our findings, we hypothesized patterns of gene expression associated with metabolic activity.

### 4.2. Negative Correlation of NPY Axis and SUVmax in Enrichment Analysis

In our enrichment analyses, the expression of genes in the NPY signaling axis and rhodopsin-like GPCRs were found to be decreased as glucose uptake indicating glycolysis increased.

NPY signaling utilizes rhodopsin-like receptors, thereby promoting inflammation [77,78] and differentiation. The role of *NPY* and its receptors in cancer is not completely understood. NPY receptors are overexpressed in different cancer entities [79]—yet studies are sometimes contradictory and suggest a very context-specific role of NPY signaling in cancers other than EwS [80,81,82].

NPY pathway expression and function in EwS has been studied as well [14,83,84,85,86,87]. *NPY* and its receptors *NPY1R* and *NPY5R* are targets of *EWS-FLI1*, and therefore upregulated in EwS [86,88]. NPY signaling was shown to foster bone metastasis in vivo. The pro-metastatic and proliferation signaling of NPY was conveyed by the receptors *NPY2R* and *NPY5R* [87,88]. However, Tilan et al. [86] showed in vitro that NPY signaling via the receptors *NPY1R* and *NPY5R* promoted cell death. A survival analysis of a publicly available EwS dataset on the R2 platform (Savola dataset, n = 44) showed significantly longer overall and event-free survival for tumors with high *NPY1R* or *NPY5R* expression. In our cohort, the expression of *NPY, NPY1R,* and *NPY5R* was negatively associated with SUVmax. We infer from this negative association that there is no uniform upregulation of these genes in all EwS tumors. Instead, the expression seems to be related to the glucose uptake of the tumor. We hypothesize that there is more *NPY* signaling promoting cell death in EwS tumors with low glucose uptake, possibly associated with neuroectodermal differentiation.

### 4.3. Spectrum of Stemness to Differentiation

Strikingly, many of our findings had a link to stemness or differentiation.

A total of 4/5 genes that were significantly associated with SUVmax in our EwS cohort have functions regarding stemness or differentiation: *FAXDC2* plays a role in megakaryocyte differentiation; *NETO2* activates tumorigenic, stemness-related signaling pathways [89,90]; *ELOVL2* is upregulated in glioma stem cells in glioblastoma, which is mediated by stem cell enhancers like *SOX2* [91]; *MYBL2* maintains an undifferentiated state of cells [92].

Furthermore, the three TFs whose targets were enriched among the genes that were positively correlated with SUVmax are associated with maintaining stemness: *RNF2* [12,93], the E2F family [94], and *TCF3* [95].

Because of these functional links described in the literature, we consider the possibility that the genes and TFs associated with stemness in other tumors or tissues are also involved in stemness in EwS, and may imply increased stemness in tumors with higher glucose uptake. This hypothesis demands further experimental validation.

Additionally, based on the findings of the NPY axis, we speculate about potential neuroectodermal and endothelial differentiation of cells at lower SUVmax. Similar findings were reported in studies on esophageal cancer [29] and lung carcinoma [96], where the authors found a correlation of high SUVmax and poorly differentiated tumors.

A stem cell-like phenotype is a basic characteristic of EwS, and maintaining stemness plays an important role [9,10,12]. Our group [12] and Sheffield et al. [19] found that EwS tumors exhibit a spectrum of stemness varying from stem-like towards mesenchymal, neuroectodermal, or endothelial differentiation. However, correlation analyses with outcome were not performed. An analysis of stemness and prognosis was conducted by Stahl et al. [11], who found a trend of high *HIF1A* expression—reflecting stemness—and shortened overall and event-free survival in EwS. This is in concordance with the general observation that stemness in cancer is predominantly associated with poor prognosis [97].

Given our findings described above, we hypothesize that the spectrum of stemness to differentiation, which was described for EwS tumors by Sheffield et al. [19], could be reflected in the SUVmax of the tumor indicating glycolytic activity. Note that we observe correlations on our study where cause and effect are unknown.

## 5. Conclusions

In our study, we characterized EwS tumors in terms of their variable glucose uptake measured as SUVmax in PET. With this, we aimed to identify novel mechanistic candidates, as SUVmax can be prognostic based on the current literature. Since EwS tumors are quite uniform at the genomic level, we assessed gene expression. Due to the low incidence of EwS, we focused on the results of enrichment analyses, as these are more robust to single false positive findings. Thus, we identified correlations between SUVmax and neuroectodermal signaling pathways or pathways downstream of stemness-related TFs. We hypothesize that stemness may be associated with increased glucose uptake. Furthermore, increased differentiation may correlate with low glucose uptake. These mechanistic candidates warrant further validation in an external cohort, as well as in experimental and clinical settings. They may eventually lead to new therapeutic options.

Our study tested the potential of a radiogenomic approach to discover novel candidates to explain the mechanisms of malignancy in EwS. Prospective data collection and validation is needed in clinical practice in the future. Thus, if we are to reap the benefits of prospective large-scale analyses, future clinical practice must be adapted accordingly.

## Figures and Tables

**Figure 1 cancers-14-05999-f001:**
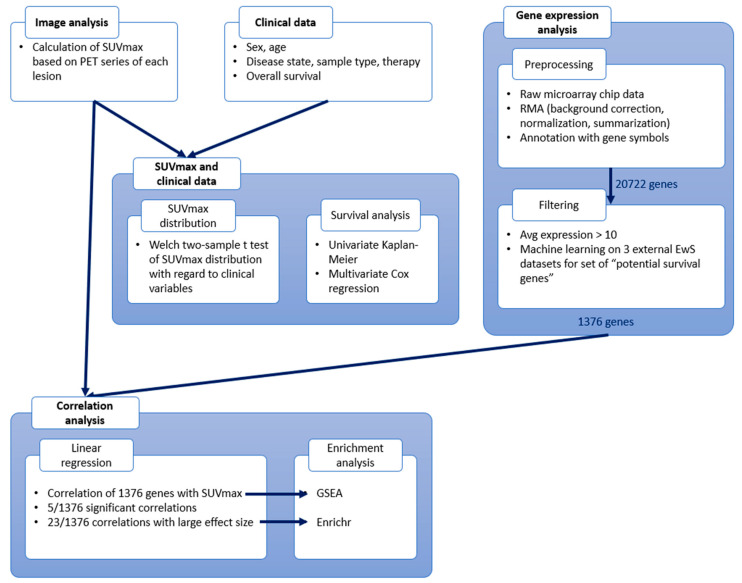
Workflow scheme. Overview of analysis steps. Positron emission tomography (PET) images are used to calculate maximal standardized uptake values (SUVmax). SUVmax distribution is analyzed with regard to clinical data, and a survival analysis is performed. Gene expression data is preprocessed and filtered, and afterwards correlated with SUVmax. The results of the correlation analysis are further annotated using enrichment analyses.

**Figure 2 cancers-14-05999-f002:**
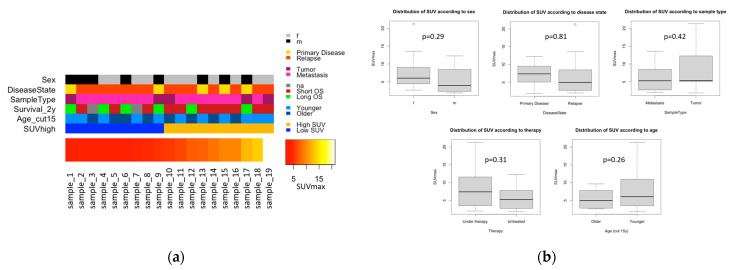
Clinical data. (**a**) 19 EwS samples ordered by increasing SUVmax having a range of 1.9 to 21.3 (bottom). On top, additional clinical information is provided about sex (female or male), disease state (primary disease or relapse), sample type (tumor or metastasis), 2-year overall survival (OS), age (≤15 years or >15 years), and categorical partitioning of the samples into high SUV or low SUV (split by median SUVmax). (**b**) Boxplots showing distribution of SUVmax values with regard to clinical variables: sex (female or male), disease state (primary disease or relapse), sample type (tumor or metastasis), therapy (under therapy or untreated), and age (≤15 years or >15 years). *p*-Values from the Welch two-sample *t* test.

**Figure 3 cancers-14-05999-f003:**
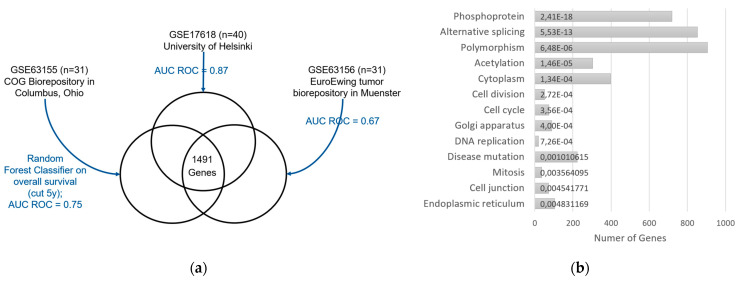
Diagram of machine learning analysis (second filtering step in gene expression analysis) in order to obtain “potential survival genes” in Ewing sarcoma (EwS). (**a**) Random forest classifiers are applied to 3 public datasets (GSE63155, GSE17618, GSE63156) in 10-fold cross-validation, obtaining an area under the receiver operating characteristic curve (ROC AUC) of 0.67 to 0.87. These models yield genes that are predictive for survival for each dataset. The intersection of these 3 gene sets contains 1491 genes, which we consider as “potential survival genes” in EwS. (**b**) DAVID functional annotation of “potential survival genes”. Thirteen terms in category UP-KEYWORDS obtained significant *p*-values (Benjamini adjusted *p* < 0.01). For each term, the number of annotated genes among the “potential survival genes” and the adjusted *p*-value is given.

**Figure 4 cancers-14-05999-f004:**
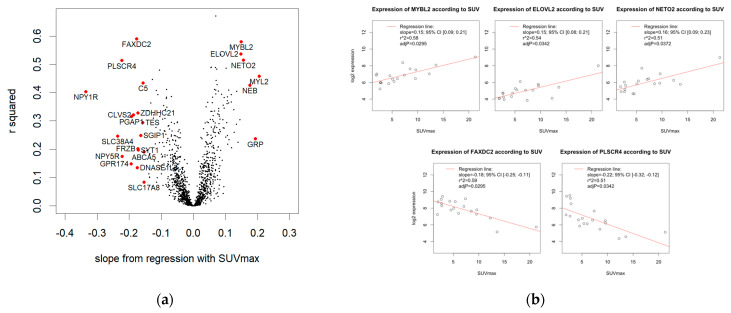
Results from linear regression modeling of SUVmax and gene expression. (**a**) Volcano plot. For each gene tested, the slope of the regression line and r^2^ (Pearson correlation) are given. Five genes are significantly correlated with SUVmax with adjusted *p*-value < 0.05 (FAXDC2, MYBL2, PLSCR4, ELOVL2, and NETO2). In addition, 23 genes show a high effect size of abs(slope) >0.146 (labelled genes). (**b**) Scatterplots of significant correlations of SUVmax and log2 gene expression. Expression of MYBL2, ELOVL2, and NETO2 is positively associated with SUVmax (top row), whereas FAXDC2 and PLSCR4 are negatively associated (bottom row).

**Figure 5 cancers-14-05999-f005:**
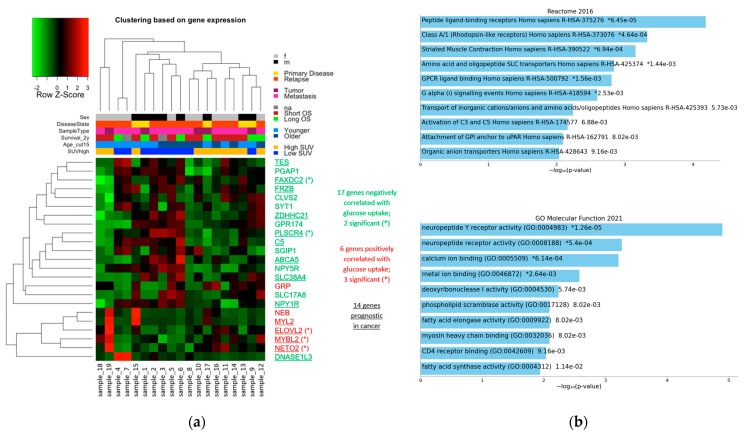
Twenty-three genes with high effect (abs(slope) > 0.146) in linear regression with SUVmax. (**a**) Heatmap depicting their expression in our cohort. On top, clinical data is provided about sex (male or female), disease state (primary disease or relapse), sample type (tumor or metastasis), 2-year OS, age (≤15 years or >15 years), and categorical partitioning of the samples into high SUV or low SUV (split by median SUVmax). (**b**) Enrichments among the 23 genes with high effect size found by Enrichr (raw *p*-values are provided, * indicates adjusted *p*-value < 0.05). Top: enrichments of the “Reactome Database”. Considering terms with adjusted *p*-value < 0.01 as significant, one term obtains a significant adjusted *p*-value (q = 0.0043): “Peptide ligand-binding receptors Homo sapiens R-HAS-375276”. Bottom: enrichments of “GO molecular function”. One term obtains a significant adjusted *p*-value (q = 0.0007): “neuropeptide Y receptor activity (GO:0004983)”.

**Figure 6 cancers-14-05999-f006:**
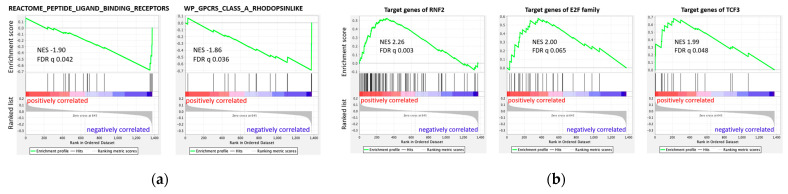
Enrichment plots of gene set enrichment analysis (GSEA) analysis on 1376 genes ordered by positive to negative correlation with SUVmax. (**a**) For canonical pathways (C2cp), 2 gene sets of rhodopsin-like receptors are enriched among genes that are negatively associated with SUVmax: “REACTOME_PEPTIDE_LIGAND_BINDING_RECEPTORS” and “WP_GPCRS_CLASS_A_RHODOPSINLIKE”. (**b**) Three transcription factors (in C3tft) show enrichment of their target genes among genes that are positively associated with SUVmax: RNF2, E2F family, TCF3.

**Table 1 cancers-14-05999-t001:** Patients’ characteristics of our EwS cohort.

		Number	Fraction
Total		19	1
Sex	Female	11	0.58
	Male	8	0.42
Disease state	Primary disease	5	0.26
	Relapse	14	0.74
Sample type	Tumor	5	0.26
	Metastasis	14	0.74
Therapy	Untreated	12	0.63
	Under therapy	7	0.37
Age at PET (all)	Number	19	1
	Range	3–31	
	Median	14	
	Mean	14.8	
Age at PET (≤15 years)	Number	11	0.58
	Range	3–15	
	Median	10	
	Mean	9.2	
Age at PET (>15 years)	Number	8	0.42
	Range	17–31	
	Median	21	
	Mean	22.5	
Imaging modality	PET-CT	15	0.79
	PET-MR	4	0.21
SUVmax (all)	Number	19	1
	Range	1.898–21.269	
	Median	5.387	
	Mean	6.819	
SUVmax low	Number	9	0.47
	Range	1.898–5.084	
	Median	2.756	
	Mean	3.207	
SUVmax high	Number	10	0.53
	Range	5.387–21.269	
	Median	9.035	
	Mean	10.07	

## Data Availability

The gene expression data generated for this study have been deposited in NCBI’s Gene Expression Omnibus and are accessible through GEO Series accession number GSE218758 (https://www.ncbi.nlm.nih.gov/geo/query/acc.cgi?acc=GSE218758) accessed on 17 October 2022. Raw PET data were generated at the Department of Nuclear Medicine, Klinikum rechts der Isar, Technische Universität München, Munich, Germany. Raw PETs are not publicly available due to patient privacy requirements. The derived data are available within the article. In addition, publicly available datasets were analyzed in this study. These data can be found in the Gene Expression Omnibus database through GEO Series accession numbers GSE63155, GSE17618, and GSE63156.

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
