# Peer review of "Correlation of Transcriptomics and FDG-PET SUVmax Indicates Reciprocal Expression of Stemness-Related Transcription Factor and Neuropeptide Signaling Pathways in Glucose Metabolism of Ewing Sarcoma"

_cancers, 2022, doi:10.3390/cancers14235999_

Round 1

Reviewer 1 Report

In this study, Prexler et al. investigated the correlation between FDG-PET imaging data and tumor expression data in a cohort of 17 Ewing sarcoma patients. For this, they first reduced the dimension of their expression dataset using previously published data for genes related to poor overall survival (OS) in Ewing sarcoma. Using this list of OS-related genes, they used different approaches to correlate expression data with SUVmax values from FDG-PET imaging. Functional annotations of these analyses highlighted pathways related to the regulation of glucose metabolism and stemness. They further highlighted 5 biomarkers correlating positively or negatively with SUVmax levels. This manuscript is well-written and provides probably one of the first radiogenomics study in this disease. Such studies are emerging in many cancers and are likely to be important for stratifying patients in future clinical trials based on these minimal invasive imaging approaches. This work is therefore very interesting but needs to address the following comments before being published:

Major comments:

 1) At this point, this study is purely correlative and does not include any validation study. Validation in an independent cohort would be great and is currently ongoing as mentioned by the author. Ewing sarcoma being a rare disease, I understand that this may take several years and may be beyond the scope of this first study. However, some in vitro experiments could be performed to validate their claims. For instance, how would the silencing of MYBL2, ELOVL2, NETO2, FAXDC2, and PLSCR4 affect glucose metabolism/stemness in Ewing sarcoma cell lines? Eventually, in vivo experiments with mouse models and FDG imaging could provide the first pieces of evidence for their claims.

2) Similarly, I would also ask to change the change Title: "Stemness-Related Transcription Factors and Neuropeptide Signaling Reciprocally Regulate Glucose Metabolism in Ewing Sarcoma" is an overstatement at this point. Indeed, there is no direct evidence for this is purely correlative at this point. This work is mostly about correlating imaging and expression data in Ewing patients, which should be somehow included as a main message in the title.

3) In figure 4b, it looks like the 3 points with SUVmax above 10 are mostly responsible for the observed correlation. Are the correlations for MYBL2, ELOVL2, NETO2, FAXDC2, and PLSCR4 still valid without these 3 "outliers"?

4) More generally, how much is the entire analysis dependent on these 3 points with a SUVmax above 10? What would be the result of their functional analyses without these 3 points?

5) In table 1, the authors indicate Relapse n= 14 and Treated n=7. How is this possible? It is highly unlikely that 7 relapse patients did not receive any treatment!

6) Since this is a single-center study, it is unclear if all imaging data were acquired on the same device or if the device was replaced over the years for instance. If this is the case, they should also evaluate this parameter and compare results with the different devices (similarly to their analyses in Fig 2b).

7) The analysis is performed with a list of genes related to Ewing OS. The methodology is clearly detailed by the authors. However, what would be the result of an unsupervised analysis with the 20'524 genes? What genes/pathways would emerge and how would MYBL2, ELOVL2, NETO2, FAXDC2, and PLSCR4 rank in this analysis?

8) lane 221: "19 samples (primary disease n=5, recurrence n=14) from 17 EwS patients". In Fig4b, how are the expression values for both patients with 2 samples each? Since these data are not independent, the average expression from these duplicate samples should probably be compared to SUVmax, unless SUVmax values for each sample are also different (may be different localization of the tumor for both samples with different SUV max?).

9) lane 260: "To obtain such survival-related genes, we used a machine learning approach on three external, public EwS datasets: GSE63155, GSE17618, GSE63156 (Figure 3a)". How many of these 1491 OS genes would be significant when investigated independently in each GSE63155, GSE17618, GSE63156 cohorts? A Venn diagram for this analysis would be nice. In addition, among these 1491 OS genes, how many are published genes related to Ewing sarcoma OS?

Minor comments:

1) Fig 4a. Please provide also a volcano plot with adjPval vs slope from regression with SUVmax as supp data.

Reviewer 2 Report

I applaud the authors for collecting an admittedly small, but impressive, set of tumor samples with correlated PET imaging from this rare cancer. I also applaud their attempts to use external datasets, filtration and enrichment to gain insight in glucose uptake in Ewing sarcoma, which has perplexed the field for many years, and take advantage of it to identify potential biomarkers and prognostic indicators.   

While I value the work the authors have done and feel it is important to share, I have concerns that study builds upon less than robust prior findings and hence overstates conclusions. 

The whole premise of this study hinges on the assumption, stated in lines 88-90 that "SUVmax may prove prognosticn in both primary and relapsed EwS". This is quite the assumption, and the literature cited (21, 22) is far from definitive. That the mean SUVmax was similar between primary and relapsed EwS cases (23) in one small cohort suggesting that what is true for primary tumors regarding SUVmax is the same for relapsed tumors, is a rather big jump as well. Furthermore, this in a way contradicts the whole notion that SUVmax correlates with resistance or "malignancy" as relapsed disease is typically more resistant and aggressive than primary disease.     

This assumption is made in passing but later on in the paper it is taken as an accepted fact that SUVmax is a correlate for outcome, which is an overstatement. 

The authors can help bolster the prognostic importance of SUVmax and outcome, and value and clinical meaningfulness to the manuscript by correlating their findings to the patient's actual outcome. Correlation to what actually happened to this manageable group of patients is surprisingly lacking. 

I understand the complexity of relaying any information regardign clinical response given that some of the patients had primary disease and some had relapsed disease, and at least a few had multiple samples, but if you are going to make the argument that SUVmax correlates with outcome and based on this you did the genomic analysis, first you must demonstrate that in your cohort SUVmax does in fact correlate with outcome. 

If you are unable to demonstrate this, or given the size of your cohort there simply is not a significant difference in outcome correlated to SUVmax, I propose restructuring the manuscript reflecting that this is an analysis to see if there is a difference in genomics as it relates to SUVmax (which is in fact the basis of this paper) and only in the discussion do you mention that past studies have suggested a correlation between SUVmax and outcomes and that while your cohort may or may not have shown this correaltion, further insight as to why Ewing sarcomas have different SUVs and what the genomic underpinnings of this might be is still of value and, as stated, a prospective large-scale analysis could validate these findings and the prognositic significance of SUVmax in Ewing sarcomas.   

Reviewer 3 Report

The general strategy is very interesting and could bring new insights but the manuscript is it written in a very confusing way. The title does not announce the real content of the article as it was not demonstrated that stemness-related transcription factors and neuropeptide signaling reciprocally regulate glucose metabolism in EwS. The authors only showed correlations between gene expression and SUVmax value. More importantly, because the methods are not clearly explained some conclusions may be wrong.

Different points have to be addressed:

L88-90: “ SUVmax may prove prognostic value in both primary and relapse EwS. Therefore, we used this parameter as a quantitative phenotype indicating metabolic tumor activity” is an example of the numerous ambiguous assertions in the text.

L126-128. Can you explain why you filtered genes with low expression?

L133-135. The patients of the external data sets were split according short- and long-term survival and the patients in the first group had OS<5years and dead. What does it mean? Patients of the long-term survival were all alive?  Were the patients selected according the dead vs alive criteria or death > vs < 5years?

L135. What criteria was considered for ambiguous patients?

Figure 2b: distribution of sex, disease state, therapy or age according to SUV should be changed for distribution of SUV according sex, disease state, therapy or age.

L233-237. Concerning the interpretation of Welch T-test: Failing to reject the null hypothesis (ns tests) is quite different from demonstrating its validity (equality of the distribution). So, you cannot rule out confounding factors and bias in subsequent analyses. This has to be corrected in the result and discussion section.

L238-245. Association of SUVmax and relapse with shorter OS should be shown even if non-significant. Why did you choose only disease state as co-variable? A clearer explanation (equations in Method section) of the model used to obtain the results would help. Attention should be payed to the interpretation of the HR. It does not allow concluding that higher SUVmax or a disease state is associated with shorter survival. Because the HR is independent on time (that why is called proportional hazards model). A HR of 5 only mean that patients of one group has 5 more chances to die at the very next moment as compared to patients of the other group and not 5 times shorter survival. A statistic that do not depend on time cannot tell anything about time. The way you expressed the results are very ambiguous.  

Line 284-285: It was reported that 645/1376 and 731/1376 genes positively and negatively correlated with SUVmax. So, all the genes that were associated with OS correlated with SUV max? But, thereafter you wrote that only 5 of these genes significantly correlated with SUV! Again, it is very confusing.

L283-285. It is not clear when you used the normalized slopes. If you normalized before analyzing positive and negative correlation of gene expression and SUVmax, you may have lost the direction of these slopes and this did not increase the interpretability but led you to false interpretation.

How confident is the strategy consisting in considering the slope value as a size effect marker? If the correlation is very loose (low R2) the error on the slope could be large.

L 290-91. How the same slope of 0.15 could indicate different doubling over SUV units.

L288 to 290. "MYBL2 showed a slope..."As stated, the reader is likely to understand that y (Gene Expression) is doubled for every 6.69 increment in x (SUVmax) which is obviously false. For instance, say that y = 0.15*x; when x=6.69, y = 1.0035 and when x = 2*6.69 = 13.38 y = 2.007, i.e. it has doubled indeed. But when x = 3*6.69 = 20.07, y = 3.0105 which is not the double of 2.007. The same apply for all corresponding statements, of course.

L297-318. Did the normalization using Z-score resulted in a normal distribution of the slope values? You have to show this to be able to deduce the probability for any slope value.

What is the meaning of disserting on the possibility to obtain a more extreme effect than those corresponding to the 5 or 23 genes? This probability is very low. Does it mean that the effects that you consider as high are not so high?

L322. How was evaluated that the 4 samples were outliers (what kind of test?). What is a slight outlier?

Did the expression of NPY and/or NPYR positively or negatively correlate with survival?

L395. you should clearly mention that you switch from the SUV-related survival genes to the entire list of 1376 survival genes and tell why.  

L406. In Methods section (L204) it was explained that GSEA was performed on the 1376 genes ranked by their correlation with SUV (from the higher slope to the lower slope, meaning from the positive correlation to the negative ones). In consequence, how could the GSEA analysis give results about the number of individual terms for low and high SUV groups that were not compared? I guess that you mean that among the genes that are positively correlated with SUV you were able to identify more terms than for the negatively correlated genes?

The FDR q values should be indicated in Figure 6.

In discussion, you reported that high NYP1R or NYP5R expression has been associated with longer OS. Have you checked the relation (positive or negative) of these genes with OS in your cohort?

The same question could be about all the genes that you highlighted in your study.

The e-learning method should be discussed. How the 1376 genes are connected to survival is not clear.

You cannot speculate about an increased stemness in tumors with high glucose up-take just because many of the genes you associated with SUV had a link with stemness or differentiation in other tumors or tissues. No term related to stemness was found in your enrichment analyses.

You cannot conclude that stemness TF and Neuropeptide signaling regulate glucose metabolism. It could be the opposite: high glucose up-take controls the stemness and signaling of tumor cells.

It is strange that you found no term or gene directly involved in metabolism.

In conclusion, the manuscript should be corrected for every ambiguous statement, the methods should be more precisely described. The results and conclusions have to be reviewed. Statistical analyses should be used and interpreted more rigorously.  

Round 2

Reviewer 1 Report

The authors have adequately answered most of my questions/comments in this revised version of the manuscript except for the in vitro experiments, which may be beyond the scope of this publication.

Author Response

We would like to thank the reviewer again for the helpful comments, so that we could improve our manuscript.

Reviewer 2 Report

I don't think adding several more references with equally shaky data related to SUVmax and prognosis in Ewings and other cancers bolsters the argument. Quality is more important than quantity. Regardless, I appreciate the authors edits to "tone down" the assertion that SUVmax is prognostic. 

I feel there is still a significant limitation to the clinical value of this study that was not addressed by the authors.

Even if we did accept that SUVmax is prognostic, how do 5 genes correlated with the SUVmax add to our understanding of the disease? It is proposed that these 5 genes may act as "biomarkers candidates for risk stratification", but what is their value for risk stratification if the SUVmax they are simply correlated with is prognostic in and of itself? It comes across as though the authors believe there is a difference between prognosis and risk stratification, and that somehow the prognostic significance of SUVmax would somehow be augmented by the identification of these potential biomarkers.  

This study would resonate much more clinically if the data of the actual patient outcomes, albeit with a small sample size, is shared with regard to its correlation with SUVmax and the 5 gene expressions. By not sharing this data, or mentioning why it is missing, a reader is left wondering if the authors are intentionally omitting it and whether it aligns with their hypotheses. 

Even if there is no correlation between real-world patient outcome and  SUVmax and the expression of these 5 genes in this dataset, it is understandable given the relatively small sample size. This would simply contribute to the author's argument that this warrants further study in a larger dataset. 

I would also ask the authors to strongly consider whether they view these 5 genes as biomarkers or potential therapeutic targets. I'm not sure biomarker is the best term to be using and I feel it detracts from the authors' argument.  

Reviewer 3 Report

Thank you for addressing every point 

Author Response

(The authors gave the same response as above.)

Round 3

Reviewer 2 Report

I appreciate the author's response to my concerns.

On the chance that I may simply not be up to date on the literature, while attended the annual international Connective Tissue Oncology Society meeting, I asked several of my colleagues who specialize in Ewing sarcoma if they use, or are even aware of, SUVmax as a prognostic tool. While they all use FDG-PET as part of staging, none of them used or was even aware of SUVmax being used as a prognostic marker in Ewing sarcoma. I am not claiming that FDG-PET has no role, nor am I arguing that SUVmax will never be used prognostically or that it isn't being studied in that role, but, at least in the United States, SUVmax is still not used clinically.     

I appreciate the authors understanding of my concerns regarding the use of the terms "biomarker" and "candidate for risk stratification".  This may be simply an issue of "track changes", but I still see "biomarker" and "candidate for risk stratification" throughout the manuscript and the terms do not appear crossed out or deleted. In some cases the term "mechanistic" is added, creating the ambiguous term "biomarker mechanistic candidate". I worry this detracts from the authors argument and does not help clarify the role proposed for these genes.   

If indeed the terms "biomarker" and "candidates for risk stratification" are eliminated from the manuscript, and the focus is more on the potential mechanistic underpinning of glucose metabolism, that I believe is the real value of this manuscript, then the Kaplan-Meyer plots can potentially be left out. Though, I believe one of the first questions that any reader will have is what effect the expression of these genes and SUVmax had on outcomes in this cohort. I feel that being transparent and presenting the data, that the authors do possess, and addressing it directly (e.g. understanding that it is a very small cohort and hence unlikely to show any significant difference) would bolster the integrity of this work, but I leave that choice to the authors.